# Vaccination with LytA, LytC, or Pce of *Streptococcus pneumoniae* Protects against Sepsis by Inducing IgGs That Activate the Complement System

**DOI:** 10.3390/vaccines9020186

**Published:** 2021-02-23

**Authors:** Bruno Corsini, Leire Aguinagalde, Susana Ruiz, Mirian Domenech, Jose Yuste

**Affiliations:** 1Spanish Pneumococcal Reference Laboratory, National Center for Microbiology, Instituto de Salud Carlos III, 28220 Madrid, Spain; brunoccorsini@gmail.com (B.C.); leire_agui@hotmail.com (L.A.); 2Centro de Investigación Biomédica en Red de Enfermedades Respiratorias (CIBERES), 28029 Madrid, Spain; sruiz@ciberes.org; 3Centro de Investigaciones Biológicas, Consejo Superior de Investigaciones Científicas, 28040 Madrid, Spain

**Keywords:** *S. pneumoniae*, vaccine protein, LytA, LytC, Pce, cell wall hydrolases, complement, phagocytosis

## Abstract

The emergence of non-vaccine serotypes of *Streptococcus pneumoniae* after the use of vaccines based in capsular polysaccharides demonstrates the need of a broader protection vaccine based in protein antigens and widely conserved. In this study, we characterized three important virulence factors of *S. pneumoniae* namely LytA, LytC, and Pce as vaccine candidates. These proteins are choline-binding proteins that belong to the cell wall hydrolases’ family. Immunization of mice with LytA, LytC, or Pce induced high titers of immunoglobulins G (IgGs) of different subclasses, with IgG1, IgG2a, and IgG2b as the predominant immunoglobulins raised. These antibodies activated the classical pathway of the complement system by increasing the recognition of C1q on the surface of pneumococcal strains of different serotypes. Consequently, the key complement component C3 recognized more efficiently these strains in the presence of specific antibodies elicited by these proteins, activating, therefore, the phagocytosis. Finally, a mouse sepsis model of infection was established, confirming that vaccination with these proteins controlled bacterial replication in the bloodstream, increasing the survival rate. Overall, these results demonstrate that LytA, LytC, and Pce can be protein antigens to be contained in a future universal vaccine against *S. pneumoniae*.

## 1. Introduction

*Streptococcus pneumoniae* is the leading bacterial pathogen of lower respiratory tract infections and a major cause of systemic disease including sepsis and meningitis, which are associated to high morbidity and mortality rates, especially in children under 5 years old and adults aged 65 years and older [1]. Prevention of invasive pneumococcal disease (IPD) is a cost-effective measure and one of the top priorities worldwide to reduce the impact of pneumococcal infections in public health [2,3,4]. Current prophylactic strategies against *S. pneumoniae* are limited to vaccines based on capsular polysaccharides (CPS). In the pediatric population, the use of pneumococcal conjugate vaccines has significantly reduced the burden of disease caused by serotypes included in these vaccines showing herd effect in adults, confirming the efficacy of these vaccines against vaccine serotypes [5,6,7]. In adults, the use of the 23-valent pneumococcal polysaccharide vaccine (PPV23) and/or the 13-valent conjugate vaccine (PCV13) in a greater extent have also been effective against IPD and pneumonia caused by serotypes contained in these vaccines [8,9,10,11,12]. However, with up to 100 different serotypes described so far on the basis of CPS, an important limitation of these vaccines is based in the extent number of serotypes of *S. pneumoniae* [13]. Serotype replacement (emergence of non-vaccine serotypes) is a common scenario for past and current anti-pneumococcal vaccines. After the use of the 7-valent pneumococcal conjugate vaccine (PCV7), the emergence of serotype 19A was promptly observed not only in children but also in adults [7,14]. This increase was rapidly contained after the introduction of PCV13, compared to countries using the 10-valent pneumococcal conjugate vaccine (PCV10) where this particular serotype remained as an important cause of IPD [12,14,15,16]. Currently, after a decade of using PCV13, the emergence of non-vaccine serotypes is of great concern worldwide with many countries reporting the rise of non-PCV13 serotypes [12,14,17,18]. To solve this problem, the discovery of pneumococcal protein antigens that can confer protection against different serotypes is increasing in the last years and may contribute to a future cocktail of proteins to be incorporated in a future universal vaccine against this important human pathogen [19,20,21,22,23,24,25,26].

One important aspect that should be considered for a vaccine candidate is the potential of the protein target to be an important virulence factor because antibodies elicited against the protein might block and/or attenuate the impact of this protein on pathogenesis [19,24,27]. In this sense, the proteins selected for our study play critical roles in different steps of the pathogenesis process. LytA is the main autolysin of the bacterium and is responsible for the release of the citotoxin pneumolysin (Ply) [28,29]. Previous studies by our group using strains lacking Ply, LytA, and both simultaneously demonstrated that LytA is essential in the evasion of the complement-mediated immunity by a Ply-independent mechanism being critical in the direct cleavage of the C3 component deposited on the bacterial surface [30]. LytC is a lytic enzyme involved in fratricide [31], biofilm formation [32], and adhesion to the nasopharygeal tract [33,34] and avoids the activation of the complement system [34]. Pce (also known as CbpE) is a protein involved in the early step process of pathogenesis by participating not only in the attachment to the nasopharynx [33] but also in the release of phosphorylcholine moieties on the bacterial surface, avoiding, therefore, the recognition by acute phase proteins such as C-reactive protein [35].

In this study, we have characterized three different pneumococcal proteins termed LytA, LytC, and Pce as potential vaccine antigens. Mice immunized with these proteins elicited IgG antibodies of different subclasses that increased the activation of C1q and C3 on the surface of pneumococcal strains of different serotypes including multidrug-resistant (MDR) strains leading to enhanced phagocytosis. As a consequence, vaccination with these proteins increased the protection rate against IPD in a mouse sepsis model of infection.

## 2. Materials and Methods

### 2.1. Bacterial Strains

Pneumococcal isolates were cultured on blood agar plates at 37 °C in 5% CO_2_ or in Todd–Hewitt broth supplemented with 0.5% yeast extract to an optical density at 580 nm (OD_580_) of 0.4 (approximately 10^8^ colony-forming units (CFU)/mL) and stored at −80 °C in single-use aliquots containing 10% glycerol. The specific details (serotype and antibiotic susceptibility) of these strains are described in Table 1.

### 2.2. Immunization Experiments in Mice with LytA, LytC, or Pce

Immunization procedures were performed in groups of BALB/c mice that were bred by Instituto de Salud Carlos III (ISCIII) animal facility. All mice used were 8–16 weeks old and, within each experiment, groups of mice were matched for age and sex. Animal experiments were performed at ISCIII in accordance with Spanish legislation (RD 1201/2005, RD 53/2013) and EU regulations (218/63/EU). All animal experiments were approved by the Animal Care and Use Committee of ISCIII (CBA PA 52_2011-v2 and PROEX 218/15). For immunization studies, pneumococcal proteins LytA, LytC, and Pce were purified, as previously described [36,37,38]. Briefly, pneumococcal proteins were produced from *Escherichia coli* BL21 (DE3) or *E. coli* DH5α containing plasmids pGL100 for LytA, pLCC14 for LytC, and pRGR12 for Pce, as previously described [36,37,38,39]. Cells were induced with 100 μM of isopropyl-thio-β-D-galactopyranoside. The incubation proceeded for 16 h at 25 °C to minimize the presence of inclusion bodies. Proteins were purified by affinity-chromatography in diethylaminoethyl cellulose equilibrated in 20 mM sodium phosphate. The purity of the isolated samples was routinely analyzed by SDS-PAGE and the proteins were stored at −20 °C. Protein concentration was determined spectrophotometrically using a Nanodrop ND-1000, measuring at 280 nm. Animals were inoculated with 20 μg of each protein mixed in a 1:1 proportion in Alum (Alhydrogel; aluminum hydroxide, InvivoGen) as the adjuvant. Groups of five mice were immunized by intraperitoneal (IP) inoculation of 200 μL of Alum alone or 200 μL of pneumococcal proteins LytA, LytC, or Pce in Alum adjuvant on days 0, 7, and 14, as previously described [27,40]. Animals were euthanized on day 21 and blood was collected from cardiac puncture and conserved at −80 °C as pooled sera for further in vitro assays. For protection experiments against sepsis, groups of at least 10 mice were immunized as previously described, followed by IP challenge on day 21 with 10^7^ CFU per mouse of serotype 23F strain representing at least the lethal dose 50 (LD50) of this strain for systemic infection. Bacterial counts were determined during the first 24 h from blood samples (6 μL per mouse) obtained from the tail vein of infected animals, as previously described [27,41]. Pneumococcal disease was monitored daily until seven days, and mice were sacrificed when they exhibited severe signs of disease.

### 2.3. Enzyme-Linked Immunosorbent Assays to Detect Ig Subclasses

Specific antibody titers present in pooled sera from five mice of each group were measured by enzyme-linked immunosorbent assays (ELISA). We used 96-well polystyrene Maxisorp plates (Nunc) coated with 0.5 µg of purified LytA, LytC, or Pce protein for 2 h at 37 °C and blocked with a phosphate buffer saline-2% bovine serum albumin (PBS-2% BSA) solution, as previously described [27]. Bound antibodies were detected by using horseradish peroxidase (HRP)-conjugated goat anti-mouse IgG, IgG1, IgG2a, IgG2b, and IgG3 (Santa Cruz) for 30 min and developed using *o*-phenylenediamine (Sigma-Aldrich) before determining the OD_492_ using a microtiter plate reader (Anthos 2020).

### 2.4. Activation of Complement-Mediated Immunity by Specific IgGs

Complement activation was determined by measuring C1q and C3 using flow cytometry assays, as described before [30,42,43,44]. Briefly, C1q binding was analyzed by incubating 5 × 10^6^ CFU of *S. pneumoniae* in 10 µL of serum (diluted to 50% in PBS) for 30 min at 37 °C using pooled sera from mice immunized with Alum alone or immunized with LytA-Alum, LytC-Alum, or Pce-Alum. Bacteria were then incubated for 1 h with rabbit anti-mouse C1q antibody (Abcam) followed by an additional incubation with fluorescein isothiocyanate (FITC)-conjugated polyclonal goat anti-rabbit IgG. Then, bacteria were fixed in 3% paraformaldehyde (PFA) and analyzed on a FACS Calibur flow cytometer (BD Biosciences) using forward and side scatter parameters to gate on at least 25,000 cells. This protocol was adapted for C3 deposition. After the 30-min opsonization at 37 °C with mice sera (as described above), bacteria were incubated for 30 min on ice with FITC-conjugated polyclonal goat antimouse C3 antibody (ICN-Cappel) diluted 1/300 in PBS. Fixation with PFA and analysis by flow cytometry were similar to those performed for C1q assays.

The results were expressed as a relative percent fluorescence index (% RFI) that measures not only the proportion of fluorescent bacteria positive for the host serum component investigated but also the intensity of fluorescence that quantifies the immune component bound [27,44,45].

### 2.5. Opsonophagocytosis (OP) Using HL-60 Cells

Phagocytosis assays were assessed using a flow cytometry assay including *S. pneumoniae* strains, described above. These strains were fluorescently labeled with 5,6-carboxyfluorescein succinimidyl ester (FAM-SE; Molecular Probes) and exposed to human HL-60 cells (CCL-240; ATCC) differentiated to granulocytes. The general conditions of the assay were based on those described previously except that clinical isolates were incubated with pooled sera from mice immunized with Alum alone or immunized with the mixture LytA-Alum, LytC-Alum, or Pce-Alum [27,46,47]. The multiplicity of infection of OP assays followed a ratio of 10 bacteria per cell. A minimum of 6000 cells was analyzed using a FACS Calibur flow cytometer. Results were expressed as a RFI, defined as the proportion of positive cells for fluorescent bacteria multiplied by the geometric mean of fluorescence intensity, which correlates with the amount of bacteria phagocytosed per cell.

### 2.6. Statistical Analysis

Data are representative of results obtained from at least three independent experiments, and each data point represents the mean and standard deviations (SD) for 3–5 replicates. Statistical analysis was performed by using two-tailed Student’s *t*-test (for two groups), whereas analysis of variance (ANOVA) followed by a Dunnett’s post hoc test was chosen for multiple comparisons. Survival was analyzed by the log-rank test. GraphPad InStat version 6.0 (GraphPad Software, San Diego, CA) was used for statistical analysis. Differences were considered statistically significant with *p* < 0.05 (*) and highly significant with *p* < 0.01 (**) and *p* < 0.001 (***).

## 3. Results

### 3.1. Antibody Response to Cell Wall Hydrolases LytA, LytC, and Pce

The immunological responses measured as IgGs of different subclasses after vaccination of mice with LytA, LytC, or Pce were assessed using ELISA specific for murine sera including total IgG, IgG1, IgG2a, IgG2b, and IgG3. Antibody levels were investigated in normal mouse serum (NMS) from naïve (non-immunized) mice or sera from mice immunized with Alum alone or with mixed LytA-Alum, LytC-Alum, or Pce-Alum. Sera from mice immunized with each of the proteins investigated elicited strong IgG levels compared to Alum alone and NMS (Figure 1A). Immunization with LytA, LytC, or Pce induced consistent levels of IgGs, predominantly of subclasses IgG1 and IgG2b followed by IgG2a and IgG3 (Figure 1). Overall, these results suggest that pneumococcal LytA, LytC, and Pce are immunogenic proteins that elicit systemic IgG antibodies of different subclasses that might be protective for IPD in terms of further activation of the host-immune response.

### 3.2. Complement Activation Mediated by Antibodies to LytA, LytC, or Pce

Recognition by C1q, the first component of the classical pathway, was investigated analyzing four different clinical isolates of *S. pneumoniae* (Figure 2). For these assays, we used pooled sera from mice immunized with Alum alone or with LytA-Alum, LytC-Alum, or Pce-Alum (Figure 2). Incubation of pneumococcal clinical isolates with sera containing specific antibodies to LytA, LytC, or Pce increased the recognition by C1q, indicating that immunization with these proteins triggers activation of the classical pathway against *S. pneumoniae* and that this effect seems to be serotype independent (Figure 2).

In addition, recognition by the key complement opsonin C3 was explored for the different clinical isolates (Figure 3). Hence, C3 deposition on the pneumococcal surface was significantly enhanced in the presence of specific antibodies to each of the proteins investigated (LytA, LytC, or Pce) (Figure 3).

Overall, these results investigating the interaction of complement components in the presence of antibodies to LytA, LytC, or PCe demonstrated that vaccination with these proteins induces complement-mediated responses against different serotypes of *S. pneumoniae* (Figure 2 and Figure 3).

### 3.3. Immunization with LytA, LytC, or Pce Induces Activation of the Opsonophagocytosis Process of S. pneumoniae Regardless of the Serotype

For OP assays, FAM-SE-labeled *S. pneumoniae* strains of the four different serotypes of this study were incubated with sera from mice immunized only with Alum or with LytA, LytC, or Pce proteins mixed with Alum. As negative control, bacteria incubated with Hank’s balanced salt solution (HBSS) were included to evaluate the phagocytosis level in the absence of serum components. In the HBSS control, phagocytosis was significantly reduced in comparison to the Alum group and markedly impaired compared to LytA, LytC, or Pce immune sera, confirming, therefore, the importance of complement components in phagocytosis mediated by antibodies to these proteins (Figure 4). The level of phagocytosis was significantly higher in the presence of antibodies to LytA, LytC, or Pce in comparison to Alum group, demonstrating that immunization with those cell wall hydrolases LytA, LytC, or Pce stimulates phagocytosis of *S. pneumoniae* regardless of the serotype.

### 3.4. Vaccination with LytA, LytC, or Pce Protects Mice against Systemic Infection by S. Pneumoniae

IPD is a devastating infectious process that, when produced by strains showing high levels of antibiotic resistance, is associated to high mortality rates. The individual protective activity of LytA, LytC, or Pce against IPD was investigated in a sepsis model of infection caused by the strain 48 (serotype 23F), which is an MDR clinical isolate of *S. pneumoniae* (Figure 5). Groups of mice were immunized with 20 µg of either LytA, LytC, or Pce and survival and bacterial levels in blood were analyzed. Vaccination with each of these proteins decreased bacterial counts in blood at 24 h in comparison to Alum group and increased survival rates against sepsis caused by the MDR strain (Figure 5). The surviving mice at day 7 did not have bacterial counts in blood, confirming that vaccination with these proteins cleared the systemic infection. These results confirmed that these proteins might be promising vaccine candidates against pneumococcal sepsis caused by antibiotic-resistant strains.

## 4. Discussion

Characterization of pneumococcal proteins as vaccine antigens is necessary in the existing epidemiological context of IPD because they can offer cheaper and affordable alternatives of broad coverage without promoting the serotype replacement phenomena that is frequently observed after implementation of current prophylactic measures based in CPS vaccines [12,14,17]. In this study, we characterized three pneumococcal proteins such as LytA, LytC, and Pce that belong to the choline-binding protein family and are classified as cell wall hydrolases because they are endogenous enzymes that specifically cleave covalent bonds of the cell wall [48]. These proteins are highly conserved and are involved in fundamental biological functions such as the synthesis of the bacterial cell wall, cell shape preservation, or daughter cell separation, playing a critical role in the irreversible effects caused by certain antibiotics such as β-lactam antibiotics [48]. In the case of LytA, although two families of LytA alleles have been described, nucleotide polymorphism does not affect the primary structure of the protein, in which perfect amino acid sequence conservation was observed [49]. LytC and Pce (CbpE) are also good examples of common proteins found in the pneumococcal diverse population. The pneumococcal gene LytC was detected in 100% of nasopharyngeal samples and was the gene with the highest level of expression (>10^4^ copies/mL) of the human nasopharynx [50], whereas Pce was found in all the clinical isolates investigated regardless of the serotype [51].

LytA, LytC, and Pce are surface-exposed proteins, which is an important feature for a vaccine protein candidate due to their physical accessibility to be recognized by opsonizing antibodies and because they can interact with important components of the immune system involved in the generation of specific and long-term production of immune responses [20]. One important aspect that increases the interest for an antigen to become a vaccine candidate is the relevance of the protein in the virulence of the microorganism. These three proteins play key roles in different phases of the pathogenesis process, although one common feature among these three proteins is that they avoid the complement-mediated immunity and phagocytosis leading to the development of pneumonia and IPD [30,34,35].

In this study, we used alhydrogel as adjuvant in the immunization of mice because this adjuvant has been approved for human use in studies evaluating pneumococcal proteins [52,53]. Vaccination with LytA, LytC, or Pce was able to induce strong IgG responses of different subclasses such as IgG1, IgG2a, and IgG2b as predominant and IgG3 in a lesser degree. This aspect is relevant from the immunogenic perspective because the current PPV23 vaccine also elicits some of these subtypes of IgGs, predominantly of the IgG2 subclass [54,55]. Moreover, these three proteins also produced antibodies similar to those elicited by PCVs. In adults, PCVs produce predominantly an IgG2 response, whereas in children the primary response is of the IgG1 subtype [55,56,57]. Overall, the wide variety of IgG antibodies observed after vaccination with LytA, LytC, or Pce suggests that these proteins could be potential antigens for a vaccine targeting the pediatric and elderly population who are the major age groups at risk for developing IPD. Production of a robust IgG response has functional consequences because IgG1 and IgG3 antibodies induce a higher level of recognition by complement components and stronger interaction with Fcγ receptors than IgG2 [58,59,60]. Hence, incubation of pneumococcal strains of four different serotypes with antibodies to LytA, LytC, or Pce increased the recognition by C1q, demonstrating that vaccination with these proteins triggers the activation of the classical pathway. This is of great relevance from the pneumococcal pathogenesis perspective as this pathway plays a critical role in host defense against *S. pneumoniae* in human and mice [42,61]. In addition, vaccination with any of these three proteins activated the opsonization of different pneumococcal strains by the key complement protein C3, which is an essential component for further activation of the phagocytosis process [42]. In this sense, we found that antibodies to LytA, LytC, or Pce increased OP by human neutrophils and this effect was independent of the CPS as antibodies elicited by any of these three proteins efficiently recognized different serotypes. This is in agreement with current commercialized pneumococcal vaccines, as opsonophagocytosis is a widely accepted method to demonstrate that antibodies elicited by these vaccines are functional [62,63]. Our results are interesting in terms of functional protection because the efficacy of the host immune system to recognize and destroy *S. pneumoniae* depends on OP by neutrophils as one of the most relevant phagocytes against this bacterium [47,64], as it can be seen in neutropenic patients who are highly susceptible to pneumococcal infection [65]. Our findings are in agreement with previous observations using LytB, which is a different cell wall hydrolase, as vaccination with this protein also activated complement-mediated phagocytosis [27].

Immunization with LytA, LytC, or Pce produced an IgG3 response that may be important against invasive disease as this immunoglobulin is highly protective against systemic pneumococcal infection [66]. Our results demonstrated that mice vaccinated with these proteins controlled bacterial replication in the systemic circulation, reducing the severity of the infection process and leading to increasing survival rates. These results for LytA and LytC as potential vaccine antigens are in agreement with previous reports showing that immunization of mice with LytA or LytC increased the survival in comparison to the placebo group, although the immunological mechanism responsible for this protective effect was not explored [28,67,68]. In addition, the immunogenicity of these two proteins was confirmed using human sera from convalescent patients suffering IPD, demonstrating the presence of IgG antibodies against LytA and LytC in these patients [20,67]. In the case of Pce, our study is the first report showing not only that this protein is immunogenic but also induces activation of the host-immune response by a complement-dependent mechanism and vaccination with this protein confers protection against sepsis.

We used an MDR clinical isolate displaying high levels of antibiotic resistance to β-lactam antibiotics and macrolides to evaluate the protective capacity of the cell wall hydrolases. The reason for using this strain was because, in the clinical practice and from an epidemiology perspective, these strains are of great concern worldwide and a serious threat for the outcome of the infection and, therefore, prevention of IPD caused by these strains is a priority [69,70,71]. The main limitation of MDR strains is that acquisition of antibiotic resistance is associated to a certain loss of virulence in mice models of infection and that may be the reason that we did not observe full lethality in the placebo group immunized with alum as adjuvant [72,73]. Overall, the results of our study suggest that LytA, LytC, and Pce are promising antigen candidates to be considered in a future universal vaccine against systemic infection caused by *S. pneumoniae*.

The use of proteins as vaccine candidates can offer certain advantages compared to traditional vaccines based in capsular polysaccharides such as broader coverage and cheaper production. However, vaccine antigens based in bacterial proteins, such as the serogroup B meningococcal vaccine, do not provide a clear impact on carriage in vaccinated children [74,75]. Cocktails of pneumococcal proteins or even combination of vaccine proteins with pneumococcal polysaccharides can provide additive or synergistic effects that may be beneficial and reduce the impact of allelic variation [19,21,22,23,24,25]. In this sense, additional studies combining cell wall hydrolases including LytB might be a promising alternative that deserves further research

## 5. Conclusions

This study demonstrated that immunization with cell wall hydrolases LytA, LytC, or Pce of *S. pneumoniae* elicited an antibody response of IgGs of different subclasses including IgG1, IgG2a, IgG2b, and IgG3. These antibodies increased complement-mediated phagocytosis through activation of the classical pathway. Vaccination with these proteins controlled bacterial replication in the bloodstream, increasing the survival against systemic infection caused by an MDR pneumococcal strain.

## Figures and Tables

**Figure 1 vaccines-09-00186-f001:**
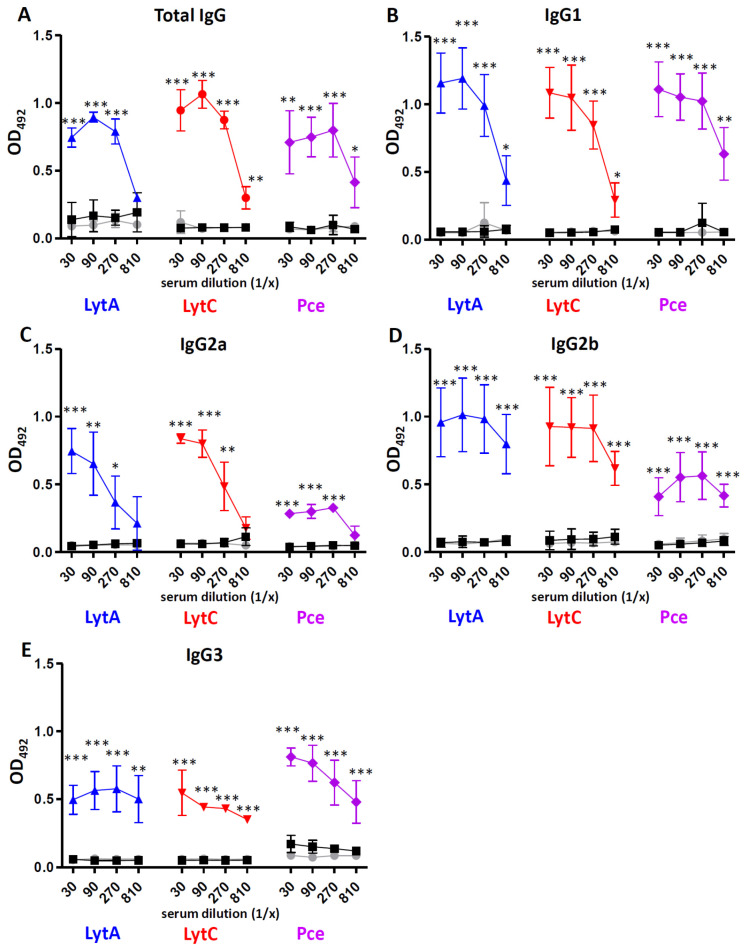
Antibody levels in sera after immunization of mice with Alum or different cell wall hydrolases (LytA, LytC, or Pce) mixed in Alum including total IgG (**A**), IgG1 (**B**), IgG2a (**C**), IgG2b (**D**), and IgG3 (**E**). Specific antibodies were measured in NMS (gray circles), pooled sera from mice immunized with Alum as adjuvant (black squares) and pooled sera from mice immunized with 20 µg of LytA-Alum (blue triangles), LytC-Alum (red circles), or Pce-Alum (purple diamonds). Error bars represent the SDs and asterisks indicate statistical significance of immunization of each protein compared to the Alum group; *****
*p* < 0.05; ** *p* < 0.01; *** *p* < 0.001.

**Figure 2 vaccines-09-00186-f002:**
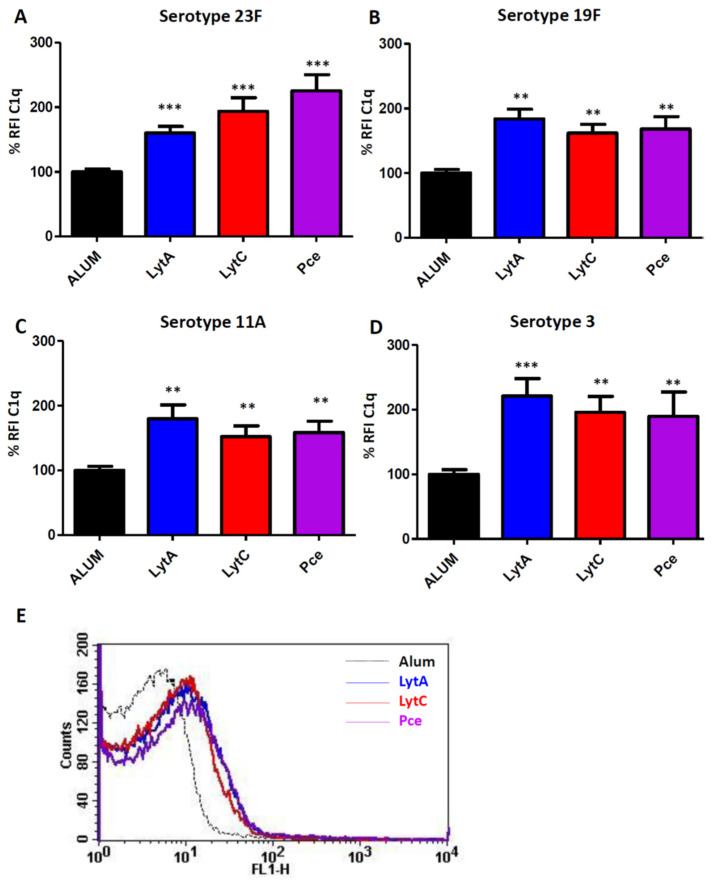
Classical pathway activation mediated by antibodies against LytA, LytC, or Pce. These assays measured deposition of mouse C1q on the surface of the corresponding bacterial strain using pooled sera from mice immunized with Alum (black bars) or with LytA-Alum (blue bars), LytC-Alum (red bars), or Pce-Alum (purple bars). (**A**) Strain 48 (serotype 23F). (**B**) Strain 69 (serotype 19F). (**C**) Strain 450 (serotype 11A). (**D**) Strain 957 (serotype 3). (**E**) Example of a flow cytometry histogram for C1q deposition on the serotype 23F strain. Results are expressed as a relative % fluorescence index (% RFI). Error bars represent the SDs and asterisks indicate statistical significance of LytA, LytC, or Pce immunization compared to the Alum group; ** *p* < 0.01; *** *p* < 0.001.

**Figure 3 vaccines-09-00186-f003:**
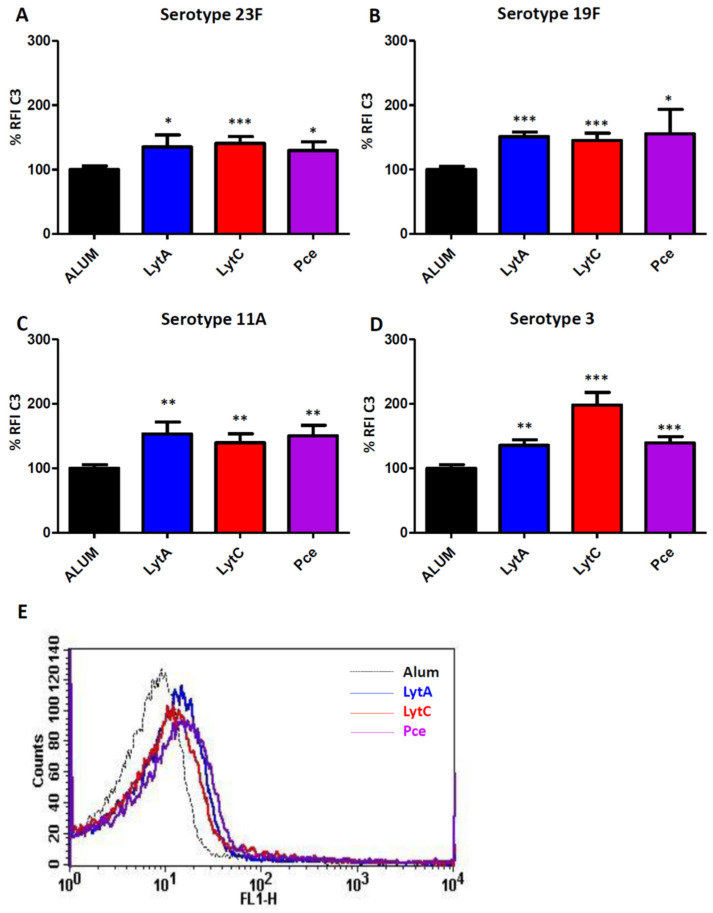
Recognition of different pneumococcal serotypes by C3 mediated by antibodies to LytA, LytC, or Pce. The different strains were exposed to pooled sera from mice immunized with Alum (black bar) or with LytA-Alum (blue bars), LytC-Alum (red bars), or Pce-Alum (purple bars). (**A**) Strain 48 (serotype 23F). (**B**) Strain 69 (serotype 19F). (**C**) Strain 450 (serotype 11A). (**D**) Strain 957 (serotype 3). (**E**) Example of a flow cytometry histogram for C1q deposition on the serotype 23F strain. Results are expressed as % RFI. Error bars represent the SDs and asterisks indicate statistical significance of LytA, LytC, or Pce immunization compared to the Alum group; * *p* < 0.05; ** *p* < 0.01; *** *p* < 0.001.

**Figure 4 vaccines-09-00186-f004:**
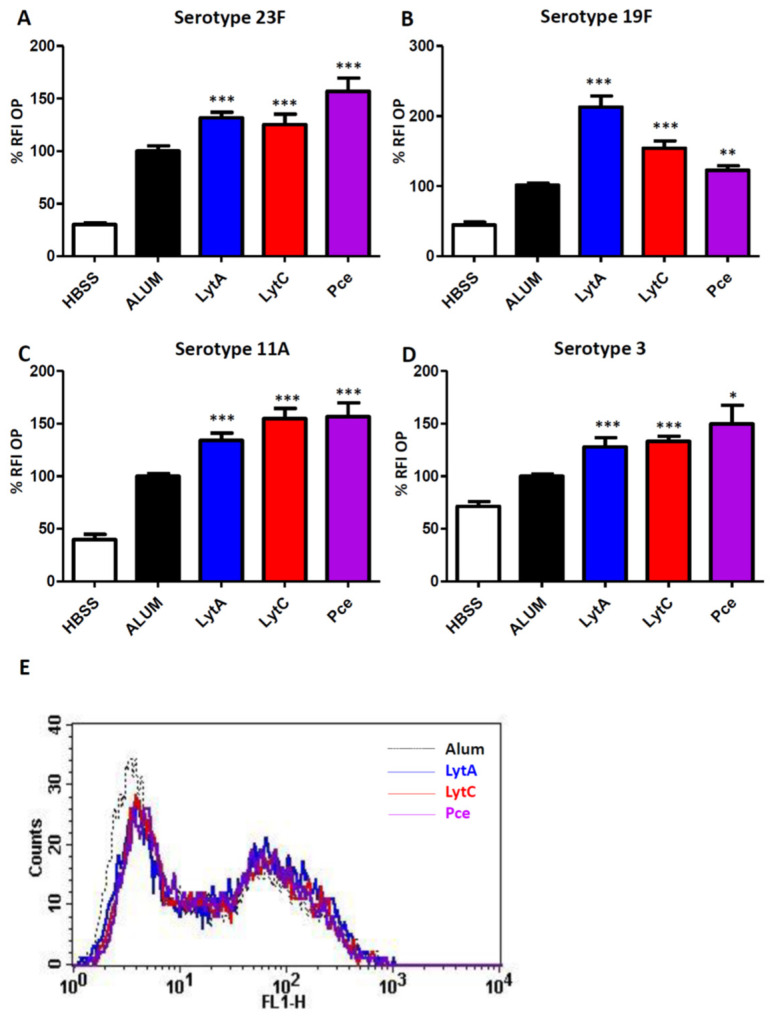
OP assays using the HL-60 neutrophil cell line and *S. pneumoniae* strains incubated with HBSS (white bar) or pooled sera from mice immunized with Alum (black bar) or with LytA-Alum (blue bars), LytC-Alum (red bars), or Pce-Alum (purple bars). (**A**) Strain 48 (serotype 23F). (**B**) Strain 69 (serotype 19F). (**C**) Strain 450 (serotype 11A). (**D**) Strain 957 (serotype 3). (**E**) Example of a flow cytometry histogram for OP of the clinical isolate of serotype 23F. Results are expressed as % RFI. Error bars represent the SDs and asterisks indicate statistical significance of LytA, LytC, or Pce immunization compared to the Alum group; * *p* < 0.05; ** *p* < 0.01; *** *p* < 0.001.

**Figure 5 vaccines-09-00186-f005:**
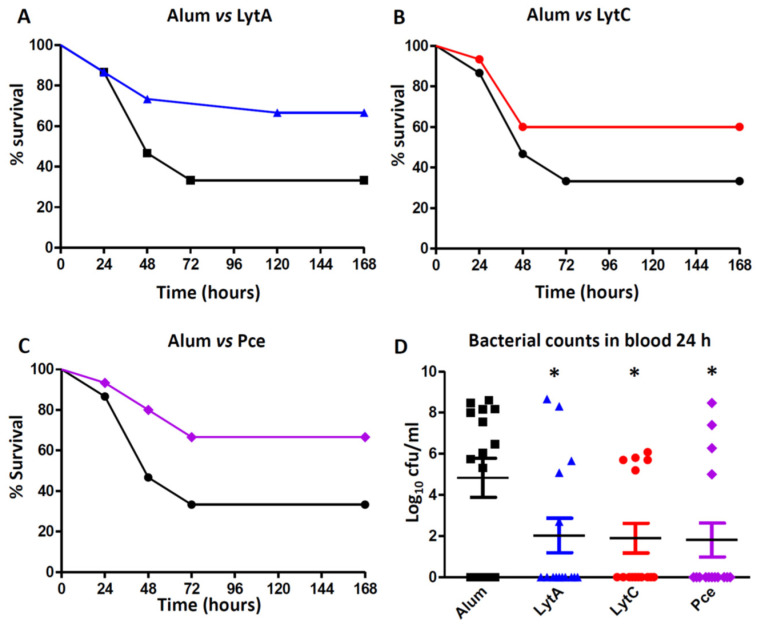
Protection mediated by LytA, LytC, or Pce against pneumococcal sepsis caused by strain 48 (serotype 23F). Mice were vaccinated with Alum or with 20 µg of the specific protein (LytA, LytC, or Pce) mixed with Alum. (**A**) Survival in mice vaccinated with Alum (black line) or LytA-Alum (blue line). (**B**) Survival in mice vaccinated with Alum (black line) or LytC-Alum (red line). (**C**) Survival in mice vaccinated with Alum (black line) or Pce-Alum (purple line). (**D**) Bacterial counts in blood at 24 h from mice immunized with Alum or proteins (LytA, LytC, or Pce) mixed with Alum. Error bars represent the SDs and asterisks indicate statistical significance of vaccination with LytA, LytC, or Pce compared to the Alum group; * *p* < 0.05.

**Table 1 vaccines-09-00186-t001:** Pneumococcal clinical isolates used in this study. Serotype is indicated as well as the minimum inhibitory concentration (MIC) (μg/mL) to different antibiotics ^1^.

Strain (Serotype)	PEN	AMX	CTX	TET	CHL	ERY	LVX
48 (23F)	8	16	8	64	4	>128	2
69 (19F)	2	2	2	4	4	>128	1
450 (11A)	2	4	0.5	0.25	4	0.12	1
957 (3)	0.015	0.015	0.015	0.5	4	0.25	1

^1^ PEN (penicillin); AMX (amoxicillin); CTX (cefotaxime); TET (tetracycline); CHL (chloramphenicol); ERY (erythromycin); LVX (levofloxacin).

## Data Availability

The data presented in this study is contained within the article.

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
