# Peer review of "Vaccination with LytA, LytC, or Pce of Streptococcus pneumoniae Protects against Sepsis by Inducing IgGs That Activate the Complement System"

_vaccines, 2021, doi:10.3390/vaccines9020186_

Round 1
Reviewer 1 Report
Authors, in this MS titled " Vaccination with LytA, LytC or Pce of S. pneumoniae protects against sepsis by inducing IgGs that activate the complement system" have evaluated the utility and efficacy of protein antigens as an alternatives/complemental antigens for vaccination. The MS is overall well written and straight forward. Following areas may be improved to increase the quality of the MS.
Major:
- Despite their citations in the methods, a brief description of preparation of antigens/proteins is needed, including their native structure maintenance.
- What strain of the bacteria used for protein antigen preparation?
- A brief description of conservation of these proteins, at least across the strains used for assessing the vaccine effects.
- Bacterial counts were provided at 24hr post-challenge, apparently before the loss of significant numbers of mice. Can authors give the bacterial counts later time points? Another way to look into this is if survived mice have any bacterial counts?
Minor grammar or typos check is required.
Author Response
Response to Reviewer 1 Comments
Major comments
Q1: Despite their citations in the methods, a brief description of preparation of antigens/proteins is needed, including their native structure maintenance.
Answer 1: We agree with the Reviewer. The proteins used in this study were prepared and purified at the Laboratory of Prof. Ernesto García at CIB-CSIC in Spain who was the group that originally purified these proteins (references 36-38 of the original version submitted). In the methods section 2.2 of the new version, we have added a brief description as requested.
Q2: What strain of the bacteria used for protein antigen preparation?
Answer 2: The S. pneumoniae strains used were the wild-type strain R6 and the autolysin deficient mutant M31 as described in references 36-38.
Q3. A brief description of conservation of these proteins, at least across the strains used for assessing the vaccine effects.
Answer 3: We have added a paragraph in the Discussion section, explaining the conservation of these proteins. Please, see page 10 of the Discussion section.
Q4. Bacterial counts were provided at 24h post-challenge, apparently before the loss of significant numbers of mice. Can authors give the bacterial counts later time points? Another way to look into this is if survived mice have any bacterial counts?
Answer 4: We also measured bacterial counts at 48 h and the findings were similar compared to 24h with lower infection level in groups vaccinated with LytA, LytC or Pce in comparison to the Alum group. The surviving mice at day 7 did not have bacterial counts in blood confirming that vaccination with these proteins clear the systemic infection. This is explained in the results section 3.4.
Minor grammar or typos check is required.
We have reviewed the manuscript to amend minor grammar or typos.

Reviewer 2 Report
Review comments
Dear Authors and editor:
I have carefully read this manuscript. Corsini et al used three virulence factors, LytA, LytC and Pce, as recombinant protein vaccine candidate against S. penumoniae. Animal studies proved the efficacy of the vaccine at mice sepsis model. Overall, this paper is very good providing insights into the development of broad-spectrum vaccines for controlling S. pneumoniae.
The following are some questions and concerns:
- The authors used non-prescribed drugs for determining the MIC against S. pneumoniae. Why?
- Which antigen best boost the appropriate antibody?
- If combining two or three antigens, will be protection rate be improved?
- What other immune components than IgG and complement were boosted by the candidate vaccines?
- Will the combination of protein antigens and polysaccharide make the protection rate higher?
- Authors should compare with other commercial pneumococcal vaccines?
- What is the limitation of this protein antigen based vaccine?
Author Response
Q1. The authors used non-prescribed drugs for determining the MIC against S. pneumoniae. Why?
Answer 1: We agree with the Reviewer that some of the antibiotics we tested such as chloramphenicol or tetracycline are not prescribed for invasive pneumococcal disease. However, as part of our duties within the Spanish Pneumococcal Reference Laboratory, we routinely test the MIC against all these antibiotics for epidemiological purposes and that is the reason that we included all of them.
Q2. Which antigen best boost the appropriate antibody?
Answer 2: The study was not created to compare the results among the different proteins, as we wanted to characterize the protective activity of each protein individually. Our results do not support a particular protein as a better antigen because results are variable depending the component measured (IgG, C1q, C3…), and even the strain investigated. Overall, the strength of our results is that each of these proteins trigger the host immune response and can be considered as a vaccine antigen itself.
Q3. If combining two or three antigens, will be protection rate be improved?
Answer 3: The main goal of the study was to evaluate the immunogenicity of each of these proteins (LytA, LytC or Pce) and the ability of these proteins to stimulate the host immune response and protect as individual proteins. There are examples of enhanced protection when combining pneumococcal proteins. This is discussed in the manuscript and we have added that additional studies combining cell wall hydrolases including LytB might be a promising altenative that deserves further research. Please, see last paragraph of the Discussion section.
Q4. What other immune components than IgG and complement were boosted by the candidate vaccines?
Answer 4: The pneumococcal antigens tested in our study elicited IgGs of different subclasses and increased the recognition by components C1q and C3. In addition, antibodies to these vaccine candidates increased the phagocytosis process against clinical isolates of four different serotypes.
Q5. Will the combination of protein antigens and polysaccharide make the protection rate higher?
Answer 5: The question raised by the Reviewer is very interesting. Several protein antigens have been combined with polysaccharides as vaccine alternatives. This point is addressed in the Discusion section of the new version submitted. Please, see last paragraph of the Discussion section.
Q6. Authors should compare with other commercial pneumococcal vaccines?
Answer 6: In the Discussion section of the manuscript, we have compared the immune response of IgGs elicited after immunization with LytA, LytC or Pce with the IgG subtypes elicited in humans with PPV23 and conjugate vaccines. Please see the paragraph in the Discussion section citing references 50-56.
To address the comment by the Reviewer in more detail, we have explained that the results with our proteins are in agreement with current commercialized pneumococcal vaccines as opsonophagocytosis is a widely accepted method to demonstrate that antibodies elicited by these vaccines are functional, including two new references (62 and 63) for PPV23 and PCV13. Please see the Discussion section of the new version submitted.
Q7. What is the limitation of this protein antigen based vaccine?
Answer 7: We agree with the reviewer that vaccines based in protein antigens also have certain limitations. In the last section of the Discussion section of the new version submitted, we have explained the possibility of allele variability when using proteins and a lower impact on nasopharyngeal carriage compared to polysaccharide conjugate vaccines in a similar way than the meningococcal group B proteins based vaccines. Please, see the last paragraph of the Discussion section in the new version submitted.
Round 2
Reviewer 1 Report
None. All comments have been addressed reasonably.
Reviewer 2 Report
Dear authors:
Thank you very much for submitting the revision. I don't have any further questions. As a result, I'd like to recommend acceptance of this current version.